# Emulsion Properties during Microencapsulation of Cannabis Oil Based on Protein and Sucrose Esters as Emulsifiers: Stability and Rheological Behavior

**DOI:** 10.3390/foods11233923

**Published:** 2022-12-05

**Authors:** Qun Zhang, Yan Shi, Zongcai Tu, Yueming Hu, Chengyan He

**Affiliations:** 1State Key Laboratory of Food Science and Technology, Nanchang University, Nanchang 330047, China; 2National R&D Branch Center for Conventional Freshwater Fish Processing, Jiangxi Normal University, Nanchang 330022, China; 3Engineering Research Center of Freshwater Fish High-Value Utilization of Jiangxi Province, Jiangxi Normal University, Nanchang 330022, China

**Keywords:** sucrose ester, protein, rheological properties, emulsion properties, oil microcapsule

## Abstract

The effects of different emulsifiers, such as soy protein isolate–sucrose ester (SPI-SE) and whey protein isolate–sucrose ester (WPI-SE), on the properties of the emulsion during the microencapsulation of cannabis oil were studied. The influence of SE concentration on the emulsion properties of the two emulsifying systems was analyzed. The results of the adsorption kinetics show that SE can decrease the interfacial tension, particle size and zeta potential of the emulsions. The results of the interfacial protein concentration show that SE could competitively replace the protein at the oil-water interface and change the strength of the interfacial film. The results of the viscoelastic properties show that the emulsion structure of the two emulsion systems results in the maximum value when the concentration of SE is 0.75% (*w/v*), and the elastic modulus (G’) of the emulsion prepared with SPI-SE is high. The viscosity results show that all emulsions show shear-thinning behavior and the curve fits well with the Ostwald–Dewaele model. The addition of SE in the emulsions of the two emulsion systems can effectively stabilize the emulsion and change the composition and strength of the oil–water interface of the emulsion. The cannabis oil microcapsules prepared with protein-SE as an emulsion system exhibit high quality.

## 1. Introduction

Functional oils have the effects of improving intestinal health [1], anti-atherosclerosis [2] and protecting the heart [3]. However, functional oils contain many unsaturated fatty acids, which undergo oxidation and lead to oil rancidity [4], thus causing some harm to human health. The shelf-life of functional oils can be extended by spray-drying microencapsulation. The preparation of the emulsion is a key step in the preparation of functional oil microcapsules, and a stable emulsion can improve the efficiency and effect of microencapsulation. High-quality microcapsules can be produced by studying the relationship between emulsion and microcapsule properties during microencapsulation.

Proteins are often used as emulsifiers to stabilize emulsions because of their excellent amphiphilic, film-forming [5] and biocompatibility [6] properties. Proteins undergo expansion and rearrangement after they are spontaneously adsorbed on the oil–water interface because of the hydrophobic interaction of non-polar side chains in the emulsion [7]. In this set-up, the hydrophilic groups are in contact with water, and the hydrophobic groups are bound to the oil phase. Consequently, the proteins are arranged directionally and orderly on the interface. As a result, the interfacial tension is reduced, and a viscoelastic interface film is formed, which can effectively stabilize the emulsion. The type, concentration and structure of proteins affect the adsorption behavior on the interface. Seta et al. [8] found that compared with β-casein, the conformation of β-lactoglobulin changes more easily at the oil–water interface, its adsorption rate is faster, and it has a greater tendency to dominate the interface. The emulsion stabilized by casein gradually changes from Newtonian thinning to shear thinning with the increase in casein concentration [9]. ManeephanKeerati-u-rai et al. [10] found that the structure of soy protein isolate changes after heating, thus decreasing its adsorption capacity at the oil–water interface. Small molecular surfactants are also commonly used as emulsifiers, and they have higher surface activity than proteins and can be quickly adsorbed on the interface to reduce interfacial tension; through charge repulsion and the Gibbs–Marangoni mechanism, they can form a tight adsorption layer to stabilize the emulsion [11]. However, small molecular surfactants cannot stabilize the emulsion continuously and effectively over a relatively long time. Although the protein can form a viscoelastic interface film outside the oil droplets, its surface activity is low. However, the film formed by small molecular surfactant does not have viscoelasticity, and the strength of the interfacial film is weak, making it easily destroyed during processing [12]. Therefore, the combination of protein and small molecular surfactants to improve the interfacial properties of emulsion is the current research focus. The addition of small molecular surfactants, such as monoglyceride and lecithin, will change the adsorption behavior of proteins at the interface and the rheological properties of the emulsion [13,14,15].

Small molecule surfactants can interact with proteins at the oil–water interface, resulting in the partial or complete replacement of proteins [16]. Sucrose ester (SE) is a small molecular surfactant with sucrose as the hydrophilic group and a fatty acid chain as the hydrophobic group. Sucrose molecules have eight free hydroxyl groups. Different hydrophobic SE can be obtained by optimizing the esterification degree and alkyl chain properties [17], and carbon chain length mainly affects the surface properties of SE [18]. SE can not only adsorb together with proteins at the oil–water interface, but also replace proteins at the oil–water interface, depending on the concentration of SE in the emulsion [19,20]. SE can change the rheological properties of the emulsion, reduce the interfacial tension and increase the viscosity of the emulsion, but excessive SE causes negative effects [21]. In conclusion, when protein and sucrose ester are used as emulsifiers, the concentration of SE can affect the properties of the emulsion. Therefore, the use of protein–sucrose ester as an emulsifier may affect the properties of the emulsion during microencapsulation, thereby affecting the quality of oil microencapsulation. However, research on this field has rarely been reported, and it is helpful to further understand the relationship between the properties of emulsion during microencapsulation and the quality of oil microcapsules.

Cannabis oil is an edible oil extracted from dried seeds of cannabis. It is rich in unsaturated fatty acids, and it contains linoleic acid and α-linolenic acid, which are necessary for the human body, in a 3:1 ratio. Cannabis oil has the effect of preventing arteriosclerosis. In this paper, soy protein isolate–sucrose ester (SPI-SE) and whey protein isolate–sucrose ester (WPI-SE) were used as emulsifiers to study the effects of different emulsifiers and their compositions on the properties of emulsions and their relationship with the properties of oil microcapsules. Subsequently, cannabis oil microcapsules were produced.

## 2. Materials and Methods

### 2.1. Materials

Soy protein isolate (SPI, containing 90.4% protein, 0% lactose, 0.9% fat, 6.8% moisture) was purchased from Linyi ShanSong Biological products Co., Ltd. (Linyi, China). whey protein isolate (WPI, containing 91.6% protein, 0% lactose, 1.3% fat, 4.4% moisture) was purchased from Zhejiang Yinuo Biotechnology Co., Ltd. (Quzhou, Chian). Sugar ester (SE) was purchased from Guangxi GaoTong Food Technology Co., Ltd. (Liuzhou, China). Cannabis oil was purchased from Guangxi Bama yishutang health and longevity Industry Co., Ltd. (Hechi, China). All other chemicals were of analytical grade. Deionized water from a water purification system (Millipore, Billerica, MA, USA) was used throughout the experiment.

### 2.2. Preparation of Emulsion

Proteins (SPI and WPI) and SE were dissolved in deionized water at a ratio of 4:0, 4:1, 4:2, 4:3, 4:4 and 4:6, in which the concentration of protein was constant at 1% (*w/v*). After stirring for 2 h, it was stored overnight at 4 °C to form an aqueous solution. Cannabis oil (10% *w/v*) was added to the aqueous phase, and a high-speed probe (IKA T18, IKA, Königswinter, Germany) was used for dispersion at 10,000 rpm for 120 s. Then, we used the homogenizer (UH-06, Yonglian Biotechnology, Shanghai, China) to homogenize for the second time at 30 MPa to form an emulsion to be tested. According to the different emulsifiers in the emulsion, the samples were named as SPI, SPI-SE, WPI and WPI-SE emulsion.

### 2.3. Determination of Particle Size

The particle size of the emulsion was measured by a Malvern laser particle size analyzer (Mastersizer 3000, Malvern, UK). The refractive index was 1.530, and the absorptivity index was 0.100. Water was used as a dispersant, and its refractive index is 1.330. The machine measured automatically three times and took the average.

### 2.4. Determination of Zeta Potential

The Zeta potential of the emulsion was determined by Malvern laser particle size analyzer (Malvern Zetasizer Nano ZSP, Malvern, UK). Before the determination, the sample was diluted to a suitable concentration, the refractive index was 1.33, and the measurement was carried out at 25 °C. The machine measured automatically three times and took the average.

### 2.5. Observation of the Microstructure of Emulsion

The microstructure of the emulsion was observed using the method of Zhang et al. [22] with slight modification. The microstructure of the emulsion was observed using a TCS SP8 confocal laser scanning microscope (Leica, Wetzlar, Germany). The emulsion samples were stained with 0.02% (*w/v*) Nile red and 0.1% (*w/v*) Nile blue, and the appropriate amount of dyeing emulsion was observed under the microscope. Nile red was excited by an argon–krypton laser at 488 nm, and Nile blue was excited by a He-Ne laser at 633 nm. The sample was observed with 100× magnification lens, and laser confocal images were obtained via LAS AF software.

### 2.6. Determination of Interfacial Protein Concentration

The interfacial protein concentration was determined using the method of Zhang et al. [11] with slight modification. Approximately 2.0 mL of fresh emulsion was poured into a centrifugal tube and centrifuged 30 min at 10,000 rpm at 20 °C. After centrifugation, the clear liquid layer was carefully removed with a syringe and filtered through a 0.22 μm filter membrane. Then, the filtrate was collected. The BCA protein concentration determination kit (Solaibao, Beijing, China) was used to determine the concentration of the emulsion protein. Bovine serum albumin was used as the standard protein. This was repeated three times for each test. The interfacial protein loading rate (**AP*%*) and interfacial protein concentration (Γ) were calculated using Equations (1) and (2) as follows:(1)$$AP\%=(c_{t}-c_{s})/c_{t}\cdot 100\%$$
(2)$$\Gamma =(c_{t}-c_{s})\cdot d_{3,2}/6\varphi$$

In the formula, Γ is the interfacial protein concentration (mg/m^2^), *C_t_* is the total concentration of emulsion protein (mg/mL), *C_s_* is the concentration of protein in the clear solution (mg/mL), *d*_3,2_ is the specific surface area of the droplet (μm), which can be obtained by the particle size tester, and *φ* is the proportion of the oil phase (10%).

### 2.7. Determination of Surface Pressure of Emulsion

The emulsion surface pressure was determined as described by Seta et al. [8] with slight modification. The solutions were prepared by dispersing 1% (*w*/*v*) protein powder and different concentration of SE (0.00–1.50%, *w*/*v*) in deionized water. Dynamic drop shape analysis was used to detect the change of the interfacial tension (*σ*) of the protein adsorbed at the oil–water interface with the adsorption time (*t*), and the OCA25 optical contact angle meter was used as the detection system. This was repeated three times for each test. The dynamic interfacial tension is expressed by the change of surface pressure π within the adsorption time *t* (Equation (3)):(3)$$\pi =\sigma_{0}-\sigma_{t}$$

In the formula, *σ*_0_ (mN/m) is the interfacial tension of the buffer solution to the oil phase without protein, and *σ_t_* (mN/m) is the interfacial tension of the sample at time *t*.

### 2.8. Shear Rheological Measurement

The viscosity of the emulsion was determined using the method of Shuang Chen et al. [23] with a slight modification. The samples were measured using an Antonpa MCR302 rheometer (Antonpa, Graz, Austria). A certain amount of sample was obtained from the sample table and measured with a flat plate probe with a diameter of 50 mm. The measuring temperature is 25 °C, and the shear rate was ranged from 0.01 to 100 s^−1^. The viscosity of the emulsion at different shear rates was determined after being stabilized for 30 s. Then the repeatability of the measurement results was verified. The viscosity curve follows the Ostwald–Dewaele model (Equation (4)) [24]:(4)$$\eta =K\cdot \gamma^{n-1}$$

In the formula, *Ƞ* is the viscosity (Pa·s), *K* is the consistency coefficient (Pa·s^n^), *γ* is the shear rate (s^−1^), and n is the flow characteristic index (dimensionless).

### 2.9. Determination of Viscoelastic Properties

The viscoelastic properties of the emulsion were determined as described by Wang et al. [25] with slight modification, and were measured using an Antonpa MCR302 rheometer (Antonpa, Graz, Austria). A certain amount of emulsion was placed on the sample stage and measured with a cone plate probe with a diameter of 50 mm (cone angle is 0.1° cone vertex). The spacing is 0.103 mm. After stabilization for 30 s, the dynamic oscillating frequency scanning mode was selected, and the sinusoidal deformation of different frequencies (0.1~10 Hz) was applied in the linear viscoelastic range. The elastic modulus (G’) and viscous modulus (G”) of the emulsion at different oscillating frequencies were measured. The repeatability of the measurement results was verified.

### 2.10. Preparation of Microcapsules of Cannabis Oil

The emulsions were dried using a centrifugal spray drying tower (MDR.P-5, Wuxi Modern Spray Drying Equipment Co., Wuxi, China) with an inlet temperature of 180 °C and outlet temperature of 85 °C. Emulsion was fed into the main chamber through a peristaltic pump and the feed flow rate was controlled by the pump rotation speed at 24 rpm. Prepared microcapsules were collected in hermetically plastic bags (light protected environment) and stored at 4 °C.

### 2.11. Observation on the Microstructure of Microcapsules

Appropriate amount of cannabis oil microcapsules was placed on the surface of the sample stage with conductive adhesive, and the microstructure of cannabis oil microcapsules was observed using an environmental scanning electron microscope (Quanta200F, Dreieich, Germany) in vacuum.

### 2.12. Determination of Entrapment Efficiency

The entrapment efficiency of the microcapsules was determined as described by Shi et al. [26] with slight modification. Approximately 2.0000 g microcapsule powder was added to 40 mL of petroleum ether small amplitude concussion 1 min, and then filtered. The filtrate was collected in flasks with constant weight, and the filter residue was extracted with 25 mL of petroleum ether. After filtration, the filtrate was collected and combined. A rotary evaporator was used to remove petroleum ether from the filtrate. The flask was placed in an oven at 105 °C and dried to a constant weight. This was repeated three times for each test. The entrapment efficiency were calculated using Equations (5) and (6):(5)$$SO\%=(m_{2}-m_{1})/m_{0}\times 100\%$$
(6)EE%=(1−SO/TO)×100%

In the formula, *m*_1_ is the initial constant weight of the round bottom flask, *m*_2_ is the weight of the round bottom flask containing the sample after constant weight, *m*_0_ is the mass of the microcapsule powder, *SO*% is the surface oil content of the microcapsule, *TO*% is the total oil content of the microcapsule, and *EE*% is entrapment efficiency of the microcapsule.

### 2.13. Determination of Water Content and Solubility

The water content and solubility of microcapsule were determined as described by Li et al. [27] with slight modification. One gram of microcapsule powder was accurately weighed and placed in an oven at 105 °C and dried to constant weight. This was repeated three times for each test. Water content (*MC*) was calculated using Equation (7):(7)$$MC(\%)=(m_{0}-m_{1})/m_{0}$$

In the formula, *m*_0_ is the weight before drying; *m*_1_ is the weight after drying.

Each 0.50 g microcapsule powder was mixed with 50 mL deionized water and stirred by a magnetic stirrer for 5 min at 25 °C. Then, centrifugation was performed at 3000 rpm for 5 min. Then, the supernatant was poured into constant weight weighing bottle and dried to a constant weight in an oven at 105 °C. This was repeated three times for each test.

### 2.14. Statistical Analysis

The data were analyzed by SPSS16.0.0 and origin 9.1. Differences between samples and treatment effects were tested using Duncan’s multiple-range test (*p* < 0.05).

## 3. Results

### 3.1. Particle Size Analysis

Table 1 shows the particle size of SPI-SE and WPI-SE emulsions at different concentrations of SE. The SE concentration is in the range of 0.25–0.75% (*w/v*), the particle size of SPI-SE emulsion decreased with the increase in SE concentration, and no significant difference was observed when the SE concentration was above 0.75% (*w/v*) (*p* > 0.05). The particle size of WPI-SE emulsion decreased with the increase in SE concentration. When the concentration of SE is low, the particle sizes of the two emulsions are relatively large, because the protein in the emulsion is not enough to cover the surface of the oil droplets, and the hydrophobic force between the oil droplets is strong, causing the aggregation of droplets [28]. At high SE concentration, the increase in SE concentration increases the amount of SE on the interface, covering a larger area of oil droplets, and thus weakening the hydrophobic interaction between oil droplets and decreasing the aggregation between droplets; subsequently, the emulsion particle size becomes smaller and more stable, possibly because of the synergistic adsorption of protein and SE at the interface [25]. When SE is excessive, the particle size of WPI-SE emulsion further decreases (*p* < 0.05), possibly because of the accumulation and diffusion of effective emulsion components at the interface [29]. Except for 0.5% (*w/v*) and 1.5% (*w/v*) SE concentration, the particle size of SPI-SE emulsion is smaller than that of the WPI-SE emulsion, indicating that SPI-SE emulsion has a stronger ability to stabilize the emulsion. At the concentration of 0.75% (*w/v*) SE, the particle size of SPI-SE emulsion reached the minimum value of 2.803 μm, indicating that the emulsion was the most stable at this SE concentration.

### 3.2. Zeta Potential Analysis

The zeta potential can reflect the stability of the emulsion [30]. In general, the larger the absolute value of the zeta potential is, the more difficult it is for the droplet aggregates, and the more stable the emulsion is [31]. Table 1 shows the effect of SPI-SE and WPI-SE emulsion on the zeta potential and pH at different concentrations of SE. The pH of all emulsions is greater than the isoelectric point of the protein contained (pI_SPI_ ≈ 4.5, pI_WPI_ ≈ 5), so the zeta potential of all emulsions is negative. The zeta potential of the two kinds of emulsions decreased with the increase of SE concentration. The addition of SE increases the pH value of the emulsions, causing changes in the surface charge of the oil droplets, which may unfold the protein structure, thus exposing the negatively charged groups buried in the protein to the outside and increasing the charge on the interface [32]. Consequently, the electrostatic interaction between the oil droplets and the difficulty of aggregation and flocculation between the oil droplets increase, making the emulsion more stable [32]. It will not screen the potential of the oil droplet surface, due to SE being a nonionic surfactant, so the potential change of the oil droplet at different pH levels comes from the potential change of the protein surface. When the concentration of SE is more than 1.00% (*w/v*), the potential of the SPI-SE emulsion is larger than that of the WPI-SE emulsion, possibly because excessive SE leads to the further expansion of the WPI structure and exposure of more charged groups, which is consistent with the results of particle size. When the concentration of SE is 0, the zeta potential of SPI emulsion is higher than that of WPI emulsion, but in the concentration range of 0.25–1.00% (*w/v*) SE concentration, the potential of emulsion containing SPI-SE is lower than that of WPI-SE, indicating that the interaction between SPI and SE is stronger than that between WPI and SE. Therefore, the stability of the SPI-SE emulsion is higher than that of WPI-SE emulsion in this concentration range.

### 3.3. Analysis of Interfacial Protein Loading Rate (AP%) and Interfacial Protein Concentration (Γ)

The interfacial protein loading rate and interfacial protein concentration can be used to predict the strength of the interface film [20]. As shown in Figure 1A, *AP*% of the two emulsions decreases with the increase in SE concentration, possibly because SE replaces proteins from the interface [9], and the SE adsorbed on the interface occupies a certain space to prevent protein re-adsorption. The replacement process may occur through the replacement mechanism, in which the surfactant is superior to the protein in terms of reducing the interfacial free energy. In this process, the protein is replaced by the surfactant from the interface. The replacement process can also occur through the solubilization mechanism, in which the surfactant binds to the hydrophobic water point of the protein and reduces the hydrophobicity of the protein. Subsequently, the solubility in the aqueous phase increases, and the sample is desorbed from the interface [33]. Figure 1A shows that the *AP*% of the SPI-SE emulsion is much higher than that of the WPI-SE emulsion, indicating that SPI is more easily adsorbed at the oil–water interface than WPI, the amount of SPI accumulated at the oil–water interface is much higher than that of WPI. Γ and *AP*% of the two emulsions had a similar trend, and Γ decreased with the increase in SE concentration. Jiang et al. [20] also observed a similar phenomenon. They found that the interface protein concentration of casein decreased with the increase in SE concentration. However, when the SE concentration is more than 1.00% (*w/v*), Γ and *AP*% have no significant difference with the increase in SE concentration (*p* > 0.05), possibly because of the competitive adsorption equilibrium between SE and protein, and SE plays a leading role at the interface. Yang et al. [34] also found that with the increase in rhamnolipid concentration, the dominant component on the oil–water interface in the emulsion changes from protein to rhamnolipid. In summary, at the same SE concentration, Γ and *AP*% of SPI-SE emulsion were higher than those of the WPI-SE emulsion, and the thickness of the interfacial film formed by SPI-SE may be larger than that of WPI-SE.

### 3.4. Surface Pressure Analysis

The addition of a surfactant can reduce the interfacial tension and reduce the free energy to promote the stability of the emulsion. The dynamic interfacial tension is expressed by the change of surface pressure with time. As shown in Figure 2A,B, when the SE concentration is 0, the value of π increases with the increase in adsorption time, indicating that the number of proteins adsorbed at the oil–water interface increases. The value of π increases rapidly within 400 s, and at this stage, protein molecules accumulate and expand rapidly at the interface, making the surface pressure increase rapidly. The rate of surface pressure decreases with time, and the value of π is basically stable after 10,800 s. This finding shows that the adsorption of protein at the oil–water interface reached equilibrium. In the later stage of adsorption, many proteins adsorbed on the interface produced stronger steric hindrance, and the high energy barrier produced by more aggregated proteins prevented other aggregated proteins from reaching the oil–water interface [22]. Therefore, the π value is relatively stable in the later stage of adsorption. As shown in Figure 2A, the addition of SE increased the π value, indicating that SE could promote the stability of the emulsion. In comparison with other SPI-SE emulsions with SE concentration, the π value of SPI-SE emulsion with 0.75% (*w/v*) SE concentration reached the maximum, indicating that the oil–water interfacial tension of the emulsion at this SE concentration is the lowest. Zou et al. [14] also observed a similar phenomenon. They found that high concentrations of lecithin can increase the interfacial tension. The slope of the π-t^1/2^ curve represents the diffusion rate (K_diff_). A linear π-t^1/2^ curve indicates that the diffusion process is controlled by emulsifier molecule [35]. The K_diff_ of SPI-SE emulsions is larger than the K_diff_ of the SPI emulsion, indicating that the addition of SE promotes the diffusion of effective components on the interface. Unlike the SPI-SE emulsion, the difference in the π value of the WPI-SE emulsion with different SE concentrations is larger, and this condition may be related to the lower interfacial protein loading and interfacial protein concentration of WPI at the oil–water interface because the emulsifying effect of small molecular surfactants is higher than that of proteins [20]. The lower interfacial protein concentration caused a great change in π value after the addition of SE [32].

### 3.5. Analysis of Viscoelastic Properties

The viscoelastic properties of the protein adsorption layer are related to the ability of the emulsion to prevent the oil droplet from coalescence, bridging flocculation and re-coalescence [12]. The G’ curve of emulsions with SE concentration is shown in Figure 3A,B. In the frequency range of 0.1–10 Hz, the G’ curve of the emulsions systems initially increased, and then decreased with the increase in SE concentration. When the SE concentration was 0.75% (*w/v*), the G’ of the emulsion was the highest. When the concentration of SE exceeded 0.75% (*w/v*), the G’ of the emulsion decreased, because in the presence of high concentration of SE, the interaction between SE and protein molecules is reduced by hydrophobic interaction [25], Moreover, considering that the emulsifying property of SE is higher than that of protein molecules [20], part of the protein is replaced by the interface film. Figure 3C,D show that the consumption angle tangent (tan θ) of SPI-SE and WPI-SE emulsions is greater than 1, indicating that the emulsions are mainly G”, and all emulsions have no gel behaviors, highlighting the viscous behavior of the liquid [11]. In the high-frequency range (1–10 Hz), the G’ of the emulsions decreased greatly, and the corresponding G” showed an upward trend to a certain extent, indicating that the original structure was destroyed by external forces. When the concentration of SE was 0.25% (*w/v*), the G’ of SPI-SE emulsion decreased by four orders of magnitude and decreased from 1.4447 Pa to 0.00013 Pa with the increase in frequency at 3–10 Hz. At the same conditions, the G’ of WPI-SE emulsion decreased from 0.2457 Pa to 0.00004 Pa, possibly because the original structure was destroyed and the droplets were broken at the high frequency, resulting in a significant decrease in G’. At a low SE concentration, the G’ of WPI-SE emulsion is one order of magnitude lower than that of the SPI-SE emulsion, indicating that the structural strength of the SPI-SE emulsion is higher than that of the WPI-SE emulsion. In the high-frequency range, the decrease in G’ of the WPI-SE emulsion is larger than that of the SPI-SE emulsion, and the initial frequency of the decrease in G’ of the WPI-SE emulsion is less than that of the SPI-SE emulsion, indicating that the structural strength of the SPI-SE emulsion is higher than that of the WPI-SE emulsion. Therefore, the SPI-SE emulsion has the ability to prevent oil droplet flocculation and re-aggregation, and it is the strongest at the SE level of 0.75% (*w/v*).

### 3.6. Shear Rheological Analysis

The study of shear rheology can effectively obtain information about the interaction between the protein and surfactant in the interfacial adsorption layer by applying a different strain, strain rate and stress [8]. Viscosity is the ability of a non-Newtonian fluid that obeys the law of exponential flow to resist shear deformation at the action of an external force. The Ƞ of emulsions decreased with the increase in shear rate, indicating typical shear-thinning behaviors. The Ƞ measurement result was subjected to curve fitting, and the curve-fitting result fits well with the Ostwald–Dewaele model [24]. The fitting result is shown in Table 2, where R^2^ is greater than 0.98, indicating that the curve-fitting result is reliable. The n values of all curves are less than 1, indicating shear thinning. The shear-thinning behaviors may be related to the rupture and deformation of emulsion oil droplets [19], or it may be caused by the hydrophobic repulsion of oil droplets and the destruction of hydrogen bonds between the protein and protein [36]. For the SPI-SE emulsion, the absolute value of n increased with the increase in SE concentration, indicating that the addition of SE made the viscosity of the emulsion more dependent on the shear rate. By contrast, the viscosity of the WPI-SE emulsion is less dependent on the shear rate. Figure 4 shows that the viscosity of both emulsions increases with the increase in SE concentration. Gomes et al. also observed a similar phenomenon. In the emulsion with WPI and Tween 80 as the emulsifier, the viscosity of the emulsion increased with the increase in the proportion of Tween 80 [33]. When the SE concentration was more than 0.75% (*w/v*), the rising rate of the viscosity decreased, which may be related to the interfacial protein load and particle size. During interfacial adsorption, protein can dominate the interface, and *AP*% has no significant difference at a high concentration of SE. Therefore, the dominant substance of the interface changes from protein to SE. When the concentration of SE is high, the change of emulsion viscosity is very small [8].

The K value of emulsions increased with the increase in SE concentration, indicating that the viscosity of the emulsion increased. This phenomenon was observed because SE can reduce the particle size of the emulsion and enhance the fluid interaction force between oil droplets [14]. However, when the concentration of SE is 0.25% (*w/v*), the K value of the SPI-SE emulsion is less than that of the SPI emulsion, and this phenomenon may be related to the particle size of the emulsion at this concentration. Generally, the smaller the particle size, the greater the viscosity of the emulsion [19]. The internal fluidity of the fluid with high viscosity is weak, making it hard for the oil droplets to gather, making the emulsion more stable. However, the viscosity of WPI-SE emulsion is lower than that of the control group at low SE concentration, and this condition may be related to the low interfacial protein load. The low interfacial protein loading indicates increased SE adsorption, and the interface film, mainly composed of small molecular surfactants, is easily destroyed with an external force [12].

### 3.7. CLSM Analysis

As shown in Figure 5, compared with SPI and WPI emulsion, the addition of SE remarkably reduced the particle size of the emulsion. The emulsion oil droplets without SE are large, the oil droplet size distribution is uneven, and the oil droplet size varies greatly. By contrast, with the addition of SE, the large particles decreased, and the distribution of the oil droplets was more uniform: on the one hand, because the electrostatic repulsion between oil droplets increasesd on the other hand, when SE is added to the emulsion, the interfacial tension and interfacial free energy decreased. Figure 5A,D show that the particle size of the WPI-SE emulsion was remarkably larger than that of the SPI-SE emulsion, which was consistent with the results of the particle size measurement. The microstructure of emulsion at other SE concentrations is shown in Appendix A.

### 3.8. Physicochemical Properties of Microcapsules

Cannabis oil microcapsules were prepared using SPI-SE and WPI-SE as emulsifiers at 0.75% (*w/v*) SE concentration. As shown in Table 3, the moisture content of the microcapsules was lower than 3%. The low moisture content is conducive to the preservation of the microcapsules, and it is different to bond into blocks and mildew. The food industry limits the moisture content of dry powder to 3–4% [37] so the microcapsules can ensure the stability of the products within the shelf life. Solubility is an important indicator of microcapsules. High solubility is needed for microcapsules to ensure their potential application in food processing. The microcapsules made in this study had higher solubility, which may be related to the high water solubility of the components in the microcapsules [38].

The main components of SPI are 7S globulin (180–210 kDa) and 11S globulin (300–360 kDa), which account for more than 70% of the total protein. The chemical structures of 7S globulin and 11S globulin are shown in Figure 6A,B. It may lead to increased interactions between proteins due to the larger interfacial protein concentration of SPI. The interaction between 7S/11S proteins will lead to protein structure relaxation [39], exposing more hydrophobic regions. Moreover, the molecular weights of 7S and 11S are larger, which is conducive to covering a larger area on the interface. In addition, at the concentration level of 0.75% SE, the particle size and potential of lotion were smaller, the viscosity of the emulsion was larger, the fluid fluidity was weaker, and the probability of oil droplet aggregation caused by contact and collision was smaller so that the emulsion can maintain good stability during the formation of microcapsules. Therefore, the microencapsulation with SPI-SE as emulsifier had a high encapsulation efficiency, smooth surface and complete structure (Figure 7A).

The main component of WPI is β-lactoglobulin (18 kDa), accounting for more than 50% of the total protein. The chemical structure of β-lactoglobulin is shown in Figure 6C. Li et al. [40] indicated that the increase in β-fold and the decrease in α-helix in β-lactoglobulin were conducive to the adsorption of the protein at the oil–water interface. We speculated that SE may cause changes in the secondary structure of the protein, making it difficult to adapt to the oil–water interface. In addition, due to the low *AP*% and small molecular weight of WPI, excessive SE attached to the surface of β-lactoglobulin and covered part of the protein-binding region [41], which was not conducive to β- lactoglobulin adhesion on the interface. Therefore, the microcapsules with WPI-SE as emulsifier had smaller particles and rough surfaces (Figure 7B).

## 4. Conclusions

The protein-SE composite emulsifier can change the related properties of the emulsion. The addition of SE to the emulsion can reduce the particle size and potential, increase the surface pressure and viscosity, and help to maintain the stability of the emulsion during microencapsulation. At the same SE concentration, the adsorption amount of SPI at the interface is higher than that of WPI, indicating that the interfacial film formed by SPI is strong. Although SE can replace some proteins to reduce the strength of the interfacial film, it can increase the G’ of the emulsion in a suitable range of SE concentration (0–0.75%), thus improving the processability of the emulsion. Adding an appropriate amount of sucrose ester is conducive to improving the stability of the emulsion, and a stable emulsion is conducive to the production of high-quality microcapsules. Therefore, adding an appropriate amount of sucrose ester may improve the quality of the microcapsules. At the level of 0.75% SE, the microcapsules prepared with SPI-SE and WPI-SE as emulsifiers have a good appearance and high entrapment efficiency, indicating that the characteristics of the emulsion affect the effect of microencapsulation to a certain extent. The results of this study provide some theoretical support for the selection of suitable emulsifiers to produce oil microcapsules with excellent performance and controlling the process of oil microencapsulation. The emulsion system studied in this paper may not be universal; we only discussed the influence of a nonionic small molecule surfactant on the properties of emulsion during the microencapsulation, and the influence of an ionic surfactant on the properties of emulsion during the microencapsulation needs further study.

## Figures and Tables

**Figure 1 foods-11-03923-f001:**
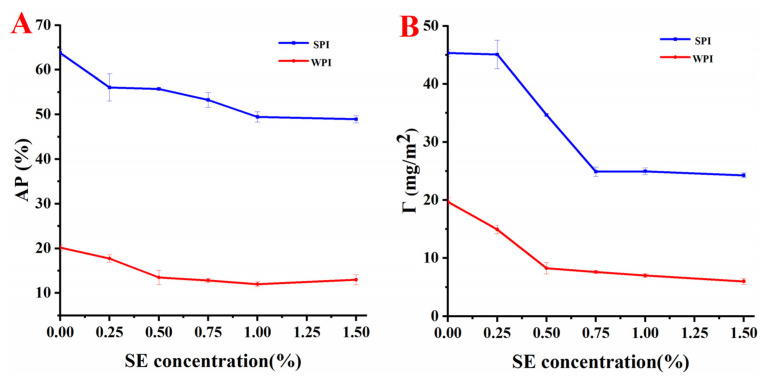
The interfacial protein loading rate (*AP*%) and interfacial protein concentration (Γ) of the emulsions at different concentrations of SE ((**A**): *AP*%; (**B**): Γ).

**Figure 2 foods-11-03923-f002:**
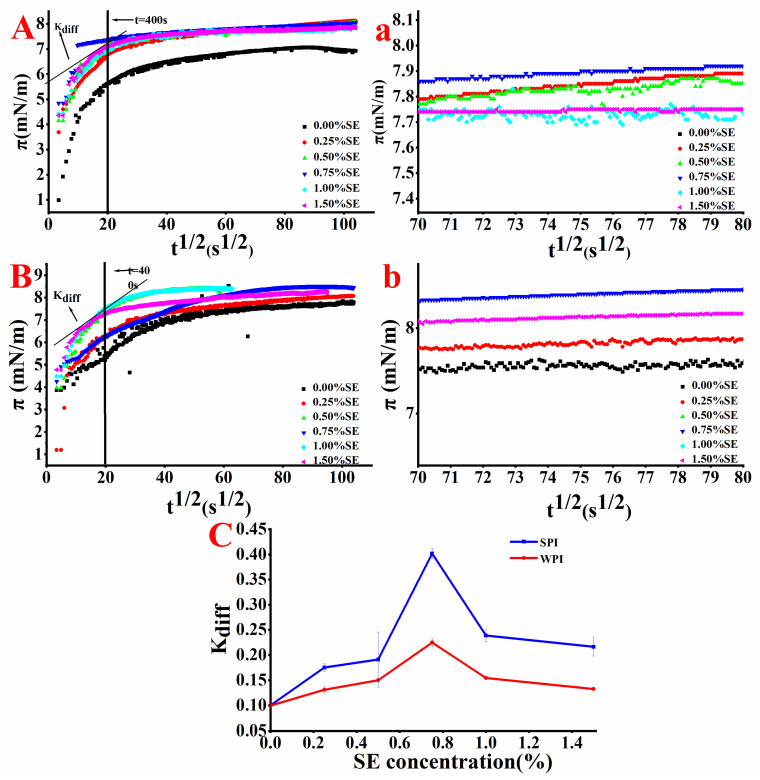
The curve of surface pressure (π) of oil-water interface of the emulsions with time square root (t^1/2^) ((**A**): SPI; (**B**): WPI); Figure (**a**) is partial enlarged view of Figure (**A**) at 70–80 s^1/2^. Figure (**b**) is partial enlarged view of Figure (**B**) at 70–80 s^1/2^. K_diff_ of emulsions at different SE concentrations (**C**).

**Figure 3 foods-11-03923-f003:**
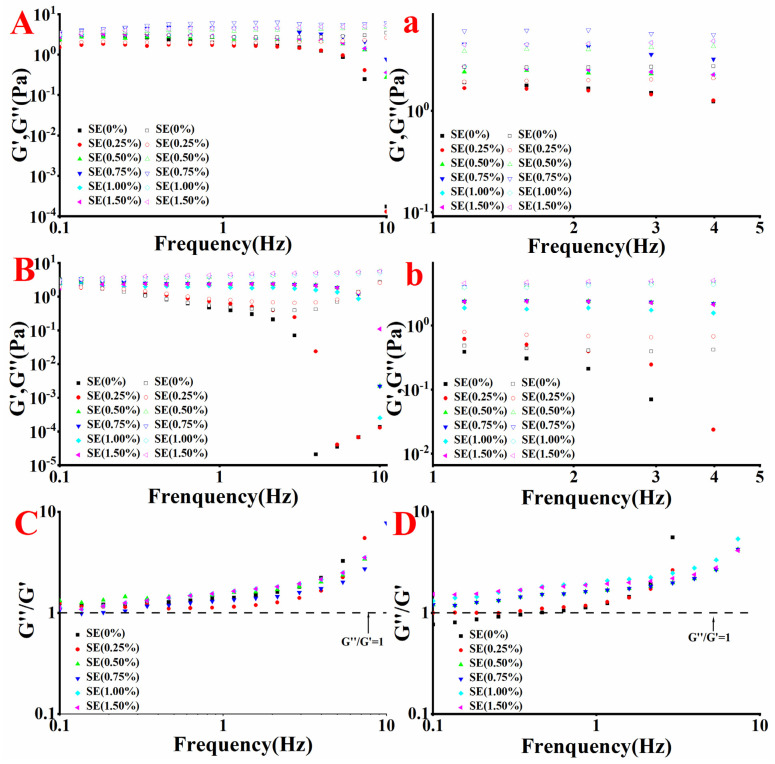
The changes of viscoelastic properties of the emulsions at different SE concentrations (**A**): SPI (the solid point is G’, the hollow point is G”); (**B**): WPI (the solid point is G’, the hollow point is G”); Figure (**a**) is partial enlarged view of Figure (**A**) at 1–5 HZ. Figure (**b**) is partial enlarged view of Figure (**B**) at 1–5 HZ. Value of the ratio of viscous modulus to elastic modulus of the emulsions at different SE concentrations ((**C**): SPI; (**D**): WPI).

**Figure 4 foods-11-03923-f004:**
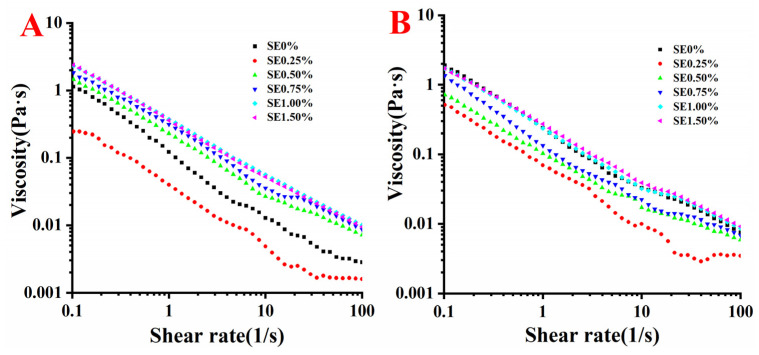
Viscosity of the emulsions at different SE concentration ((**A**): SPI; (**B**): WPI).

**Figure 5 foods-11-03923-f005:**
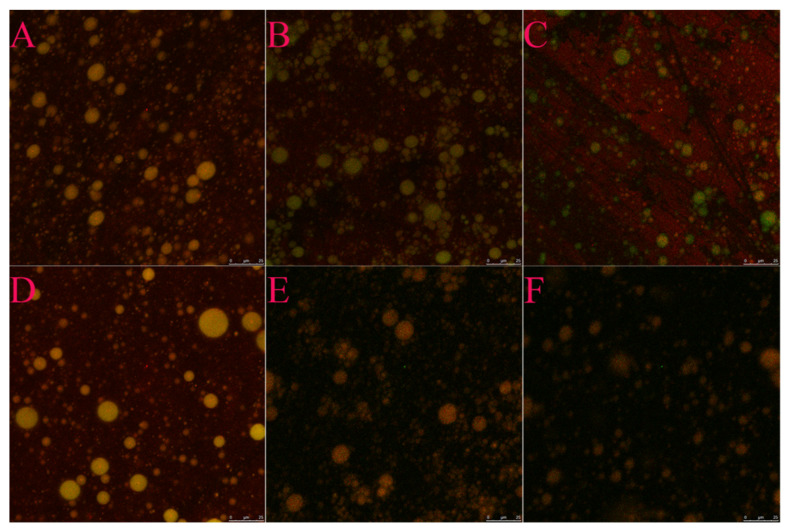
Microstructure of emulsion at different concentrations of SE. ((**A**–**C**) represent the microstructure of SPI-SE emulsion (SE0%, 0.75%, 1.5% *w/v*); (**D**–**F**) represent the microstructure of WPI-SE emulsion (SE0%, 0.75%, 1.5% *w/v*)).

**Figure 6 foods-11-03923-f006:**
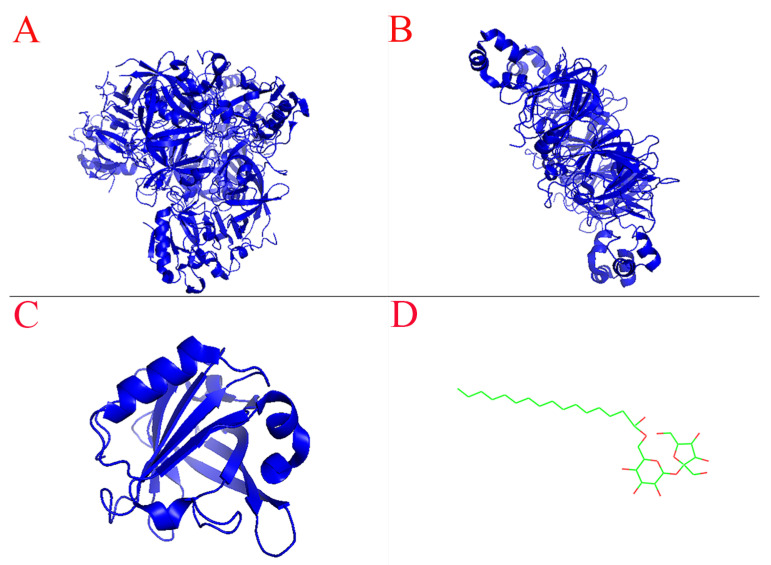
Chemical structure of 7S globulin, 11S globulin, β-lactoglobulin and sucrose ester. ((**A**), 7S globulin; (**B**), 11S globulin; (**C**), β-lactoglobulin; (**D**), sucrose ester) The chemical structures of 7S globulin (ID: 3AUP), 11S globulin (ID: 1OD5) and BLG (ID: 5IO5) were obtained from the PDB database, and the structures of SE (ID: 20003730) obtained from chemspider were visualized in PyMOL software.

**Figure 7 foods-11-03923-f007:**
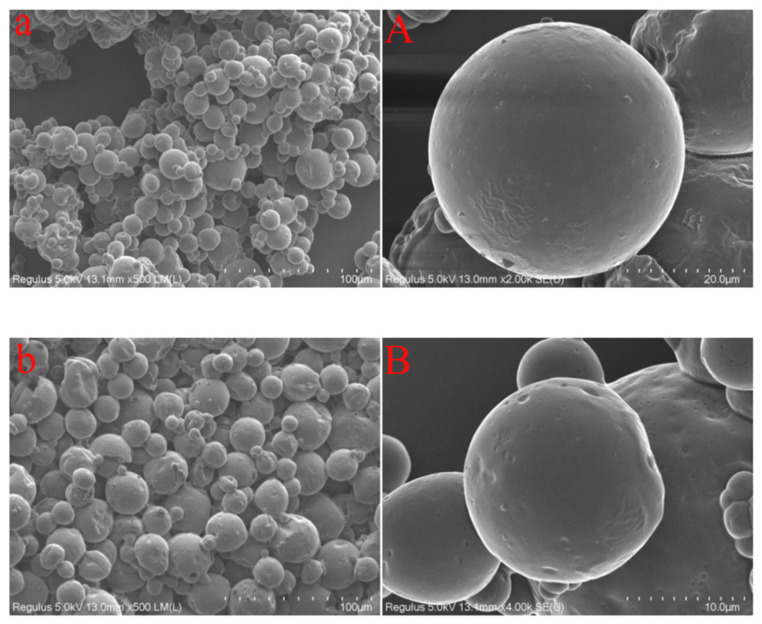
Microstructure of microcapsule of cannabis oil ((**a**,**A**):SPI-SE; (**b,B**): WPI-SE).

**Table 1 foods-11-03923-t001:** Particle size and Zeta potential of emulsion at different concentrations of SE.

	Particle Size (μm)		Zeta Potential (mV)		pH	
SE	SPI	WPI	SPI	WPI	SPI	WPI
0%	4.26 ± 0.41 ^b^	5.85 ± 0.16 ^a^	−41.03 ± 2.76 ^a^	−45.63 ± 1.02 ^a^	7.41 ± 0.03 ^a^	7.98 ± 0.03 ^a^
0.25%	4.82 ± 0.33 ^a^	5.05 ± 0.05 ^b^	−57.63 ± 1.59 ^b^	−46.87 ± 0.95 ^a^	7.46 ± 0.01 ^b^	8.45 ± 0.01 ^b^
0.50%	3.73 ± 0.22 ^c^	3.67 ± 0.29 ^c^	−60.60 ± 1.00 ^c^	−52.07 ± 2.02 ^b^	7.57 ± 0.02 ^c^	8.81 ± 0.02 ^c^
0.75%	2.80 ± 0.16 ^d^	3.57 ± 0.12 ^c^	−68.17 ± 0.29 ^d^	−54.93 ± 0.97 ^c^	7.75 ± 0.02 ^d^	9.00 ± 0.02 ^d^
1.00%	3.09 ± 0.06 ^d^	3.51 ± 0.08 ^c^	−68.57 ± 0.35 ^d^	−61.50 ± 0.27 ^d^	7.93 ± 0.02 ^e^	9.10 ± 0.02 ^e^
1.50%	2.98 ± 0.30 ^d^	2.74 ± 0.18 ^d^	−69.47 ± 2.15 ^d^	−73.00 ± 0.69 ^e^	8.19 ± 0.02 ^f^	9.31 ± 0.02 ^f^

Different letters (a–f) indicate significant difference at *p* < 0.05. Soy protein isolate (SPI); sucrose ester (SE); whey protein isolate (WPI).

**Table 2 foods-11-03923-t002:** Parameters of Ostwald–Dewaele model fitting (K: slope; n: dimensionless; R^2^: variance).

	SPI			WPI		
SE	K	n	R^2^	K	n	R^2^
0.00%	0.14 ± 0.00	0.06 ± 0.01	0.99	0.26 ± 0.00	0.12 ± 0.01	0.99
0.25%	0.05 ± 0.00	0.21 ± 0.02	0.98	0.07 ± 0.00	0.13 ± 0.01	0.99
0.50%	0.24 ± 0.00	0.20 ± 0.01	0.99	0.10 ± 0.00	0.15 ± 0.01	0.99
0.75%	0.31 ± 0.00	0.22 ± 0.01	0.99	0.14 ± 0.00	0.00 ± 0.01	0.99
1.00%	0.38 ± 0.00	0.20 ± 0.00	0.99	0.25 ± 0.00	0.15 ± 0.01	0.99
1.50%	0.37 ± 0.00	0.17 ± 0.00	0.99	0.29 ± 0.00	0.20 ± 0.00	0.99

**Table 3 foods-11-03923-t003:** Entrapment efficiency, surface oil content, moisture content and solubility of microencapsulation.

	SPI-SE	WPI-SE
Surface oil content (g/g)	0.56 ± 0.08 ^b^	0.83 ± 0.00 ^a^
Encapsulation efficiency (%)	81.33 ± 2.67 ^b^	72.40 ± 0.15 ^a^
Moisture content (%)	2.38 ± 0.12 ^b^	2.02 ± 0.20 ^a^
Solubility (%)	97.72 ± 1.92 ^a^	96.65 ± 1.51 ^a^

Different letters (a,b) indicate significant difference at *p* < 0.05.

## Data Availability

Data is contained with the article.

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
