# Peer review of "Emulsion Properties during Microencapsulation of Cannabis Oil Based on Protein and Sucrose Esters as Emulsifiers: Stability and Rheological Behavior"

_foods, 2022, doi:10.3390/foods11233923_

Round 1
Reviewer 1 Report
This is an interesting manuscript describing cannabis oil incorporation into microcapsules made from wheat or soy protein and sucrose esters as a surfactant, also rheological properties of the obtained materials. Overall, the observed findings are supported by experimental results.
Some flaws are mentioned below
-manuscript structure needs to be changed/revised. There is no section with conclusions. Also, the section on Discussion is very short, so basically, can this section can rather be considered by Authors as Conclusions?
-Figures quality needs to be improved. Please eliminate typo in Fig 1, improve quality (points are too crowded) of 2A and 2B. Figure 3 is hardly readable. Points are too small. Some points are invisible (again, crowded, modify y-axis?).
-Please eliminate excessive digits after decimal point in Table 1 and Table 2. Please adjust this number according the corresponding instrumentation capability.
-Please clarify % throughout the manuscript. Is this always wt.%? wt/v%? Something else?
-Please include chemical structures for both proteins and sucrose esters. It is important, since Authors discuss intermolecular interactions between proteins and SE as well as Authors quantitatively compare WPI and SPI performance in their experimental systems.
Finally, it looks like Authors applied same ratios of SPI and WPI in water as well as same concentration 1% (w/v). However, both proteins are biopolymers having different molecular weight, thus different number of monomeric units (amino acids) in protein macromolecules. Seems like Authors disregard the latter fact. What I am saying is that using proteins at the same concentration and same ratio does not provide a sufficient evidence for comparing the resulted material performance. This is because number of biomacromolecules in solution of WPI and SPI (assuming difference in molecular weight) is different so, in fact, hard to compare WPI and SPI performance. Authors need to address this in the revised version of the manuscript, at least provide some information about each protein molecular weight and consider how difference in molecular weight can impact the properties, in particular size of microcapsules, and, respectively, entrapment efficiency.
Author Response
请参阅附件。

Reviewer 2 Report
Manuscript ID: Foods-1988388
The article entitled: “Emulsion properties during microencapsulation of cannabis oil based on protein and sucrose esters as emulsifiers: stability and rheological behavior”.
In this paper soy protein isolate-sucrose ester and whey protein isolate-sucrose ester were used as emulsifiers to study the effects of different emulsifiers and their compositions on the properties of emulsions and their relationship with the properties of oil microcapsules. Subsequently, cannabis oil microcapsules were produced.
The article is compact and logical. From a methodological point of view, the article uses measurement techniques appropriate to the assumed purpose of the research.
Title
The title corresponds to the content of the article.
Abstract
The abstract includes the aim of the study, methods used in the experiment and contain the principal results and conclusions.
Introduction
Introduction describes the matter of the experiment and determines the examined problem. The authors correctly described the importance of research results. The cited literature refers to the subject of the analyzed problem.
Methods
The data is well collected. In the methods, more details need to be provided (below I have questions). The sampling is appropriate and adequately described. Statistical analysis of measurement results has been used.
2.8. Shear rheological measurement
In the tests, a plate-plate measuring system was used. What was the size of the gap?
2.10. Preparation of microcapsules of cannabis oil
Spray drying was carried out in a centrifugal spray drying tower. At what temperature was drying carried out?
Results
Results is very extensive while Discussion is very modestly marked. I propose to combine Results with Discussion, and supplement this part of the article with comparisons with literature.
Conclusions should be marked separately. They should be developed more.
Language
The article is correctly written. English language and style are minor spell check required.
Reviewer 3 Report
The article "Emulsion properties during microencapsulation of cannabis oil based on protein and sucrose esters as emulsifiers: stability and rheological behavior" is well designed and written. Few minor corrections have been given below and have also been highlighted in the manuscript
Introduction
Line 47, 73, 77, 117, 125, 139, 159: check the reference pattern
Line 66-81: reduce the content make it more specific and concise
Include the aim of the research
Line 109:
Line 152 rewrite
Line 155: check the equation no
Line 171: how was it stored
Line 185: check the equation no
Result: include similar/dissimilar results of studies done by other authors also
Line 396: Is it discussion or conclusion?

Round 2
Reviewer 1 Report
Manuscript was revised.
Author Response
We thank you for their constructive comments and suggestions on our manuscript.